# Impact of HPV mRNA types 16, 18, 45 detection on the risk of CIN3+ in young women with normal cervical cytology

**Khalid Al-Shibli**[1], **Hiba Abdul Latif Mohammed**[2], **Ramona Maurseth**[1], **Mikkel Fostervold**[1], **Sebastian Werner**[1], **Sveinung Wergeland Sørbye**[3]*

**1** Department of Pathology, Nordlandssykehuset HF, Bodø, Norway, **2** Department of Gynecology, Nordlandssykehuset HF, Bodø, Norway, **3** Department of Clinical Pathology, University Hospital of North Norway, Tromsø, Norway

* sveinung.sorbye@unn.no

**Data Availability Statement:** All relevant data are within the manuscript and its Supporting information files.

## Abstract

### Background

Despite a well-established cervical cancer (CC) screening program in Norway, the incidence of CC in young women is increasing, peaking at 35 years of age. 25 percent of all women diagnosed with CC had normal cytology within 3 years prior to cancer diagnosis, addressing the need to improve the screening programme to further reduce cancer incidences missed by cytology.

### Objective

We wanted to investigate the detection rate of CIN3+ in women 25–39 years with normal cytology by using a 3-type HPV mRNA test as a targeted quality assurance measure. The control group is women with normal cytology.

### Methods

During 2014–2017, samples from 13,021 women 25–39 years of age attending cervical cancer screening were analysed at Nordlandssykehuset, Bodø, Norway, including 1,896 women with normal cytology and HPV mRNA test (intervention group), and 11,125 women with cytology only (control group). The HPV mRNA testing was performed using a 3-type HPV E6/E7 mRNA test (PreTect SEE; direct genotyping 16, 18 and 45). The women were followed-up according to national guidelines throughout December 2021.

### Results

Of the 13,021 women, 429 women (3.3%) had CIN3+ confirmed by biopsy in the follow-up, including 13 cases of invasive cervical cancer. Of the 1,896 women with normal cytology and HPV mRNA test (intervention group), 49 women (2.6%) had a positive test. The risks of CIN3+ among women with either a positive or negative HPV mRNA test were 28.6% (14/49) and 0.8% (14/1847). None of the women in the intervention group developed cervical cancer during follow-up. Of the 11,125 women with cytology only (control group), 712 women

**Funding:** The author(s) received no specific funding for this work.

**Competing interests:** The authors have declared that no competing interests exist.

(6.4%) had abnormal cytology (ASC-US+). The risks of CIN3+ among women with abnormal and normal cytology were 17.7% (126/712) and 2.6% (275/10,413).

## Conclusion

By testing women 25–39 years of age with a normal cytology result using a specific 3-type HPV mRNA test, an increase in screening programme sensitivity can be achieved without an excessive additional workload. Women with normal cytology and a negative HPV mRNA test have a very low risk of cervical cancer.

## Introduction

Human papillomavirus (HPV) is the established cause of cervical cancer [1]. Therefore, HPV testing is performed for the early detection of cervical disease, both as a reflex test following cytology or as a primary screening method [2]. While it is proven that HPV-DNA testing is more sensitive for the detection of cervical dysplasia compared to cytology, its specificity for identifying women with high-grade lesions and cancer (CIN3+) is lower, especially in women under 30 years of age, largely due to the prevalence of transient, clinically benign infections [3, 4].

In Norway, cytological high-grade lesions are detected in 1.0–1.2% of the population in each screening round of the national cervical cancer screening programme [5]. The major challenge in any cervical cancer screening programme is the management of minor cervical lesions such as ASC-US and LSIL [6]. By the end of 2022, Norway aims to complete implementation of HPV-DNA primary screening for women 34–69 years of age, to improve screening sensitivity. Based on the high prevalence of HPV-infections among young women [7], cytology will remain the primary screening tool for women aged 25–33 years old. In 2019, a total of 306 cervical cancers were diagnosed among the screening population (25–69 years), where 120 (39.2%) of the cases were diagnosed in young women 25–39 years of age. Among those, 74.2% (89/120) had followed the screening programme by having a cytological sample taken within 10 years prior to diagnosis. As many as 53.9% (48/89) had a negative cytology reading before the cancer diagnosis [5]. The re-evaluation of previous cytology samples from women with cervical cancer despite attending screening often reveals cell abnormalities that were misinterpreted or overlooked the first time [8, 9]. Based on the subjectivity of cytology readings, diagnosis may vary between personnel and laboratories, and the low sensitivity remains an issue for improved prevention [10–13]. Younger women are screened using cytology and given the high number of precancerous lesions in this group, an HPV mRNA reflex test following a negative cytology result could be used as a quality assurance measure to increase screening sensitivity [14].

The HPV genome is divided into three major regions (early genes, late genes, and an upstream regulatory region). The major transforming activity of high-risk HPV is shown to be caused by the E6 and E7 oncoproteins, which interfere with the regulators of the host cell cycle and control of transcription [15, 16]. Therefore, using a test which is directed towards the detection of E6 and E7 HPV mRNA transcripts may increase the specificity of HPV testing [17, 18]. By further targeting only the three HPV types (16, 18, 45) shown to be the most prevalent types identified in about 90% of all cases of cervical cancers in young women below 40 years of age [19–21], it might be a cost-effective test to identify the women with high risk for future abnormalities among women with normal cytology.

The aim of this study was to evaluate the performance of a 3-type HPV mRNA test used as quality assurance of cytology negative women 25–39 years, by the means of test positivity rate

and risk of cervical intraepithelial neoplasia (CIN3+) compared to women with normal cytology without HPV-test.

## Material and methods

The Department of Pathology at Nordlandssykehuset-Bodø receives about 19,000 cervical cytology samples from women in Nordland County attending the national screening programme every year. During 2014–2017, a total of 41,007 women 25–69 years of age attended screening, including 13,914 women 25–39 years. After exclusion of women with previous high-grade lesions (CIN2+) and women with invalid index cytology, the final study population consisted of 13,021 women including 1,896 women with normal cytology and HPV mRNA test (intervention group), and 11,125 women with cytology only (control group), Fig 1.

All specimens were received in PreservCyt solution (Hologic, Marlborough, USA), a methanol-based preservative and processed by ThinPrep 2000 system (Cytyc Corporation, Marlborough, USA) prior to cytological examination to screen for abnormal/dysplastic cells and reported according to Norwegian guidelines.

### Classification

The department used the Bethesda system for classification of cervical cytology [22] and the WHO histological classification of tumours in cervical biopsies [23].

### HPV mRNA test

The HPV mRNA test used was PreTect SEE (PreTect AS, Klokkarstua, Norway), a qualitative assay utilising NASBA technology (targeting full length E6/E7 transcripts) with direct genotyping of amplified mRNA using molecular beacons corresponding to the HPV types 16, 18, and 45, including an intrinsic sample control ensuring sample adequacy. Artificial oligonucleotides corresponding to the viral mRNA were used as positive controls. Negative controls consisted of RNase-free water and were included in each run, according to the manufacturer's instructions.

Total nucleic acids were isolated from 1 ml of the leftover LBC material and analysed with PreTect SEE according to the manufacturer's instructions. A detailed laboratory manual is available at protocols.io [24].

### Follow-up and outcomes

Index cytology from women with a positive HPV mRNA test were re-evaluated by a second cytology technician (re-examination of the existing slide) and revised cytological diagnoses were confirmed by a pathologist. All women were followed-up in line with the Norwegian national guidelines until December 2021. In the study period, the Norwegian cervical cancer screening programme recommended women aged 25–69 to be screened with cervical cytology every three years, though some women choose to have annual screening outside of the official programme. Women with high-grade cytology were recommended colposcopy and biopsy, while women with low-grade cytology were triaged by a 14-type HPV-test. Women with normal cytology and a positive HPV-test were recommended to be re-screened after 12 months. Women with repeated positive HPV-test (HPV pos. x 2) were recommended colposcopy and biopsy. We used histologically confirmed CIN3+ as the study endpoint.

P16(INK4a) (Roche mtm laboratories AG) was used to conclude the histology diagnosis upon cellular uncertainties. If there were any discrepancies between the biopsy and treatment histology, the most severe histology was used as endpoint.

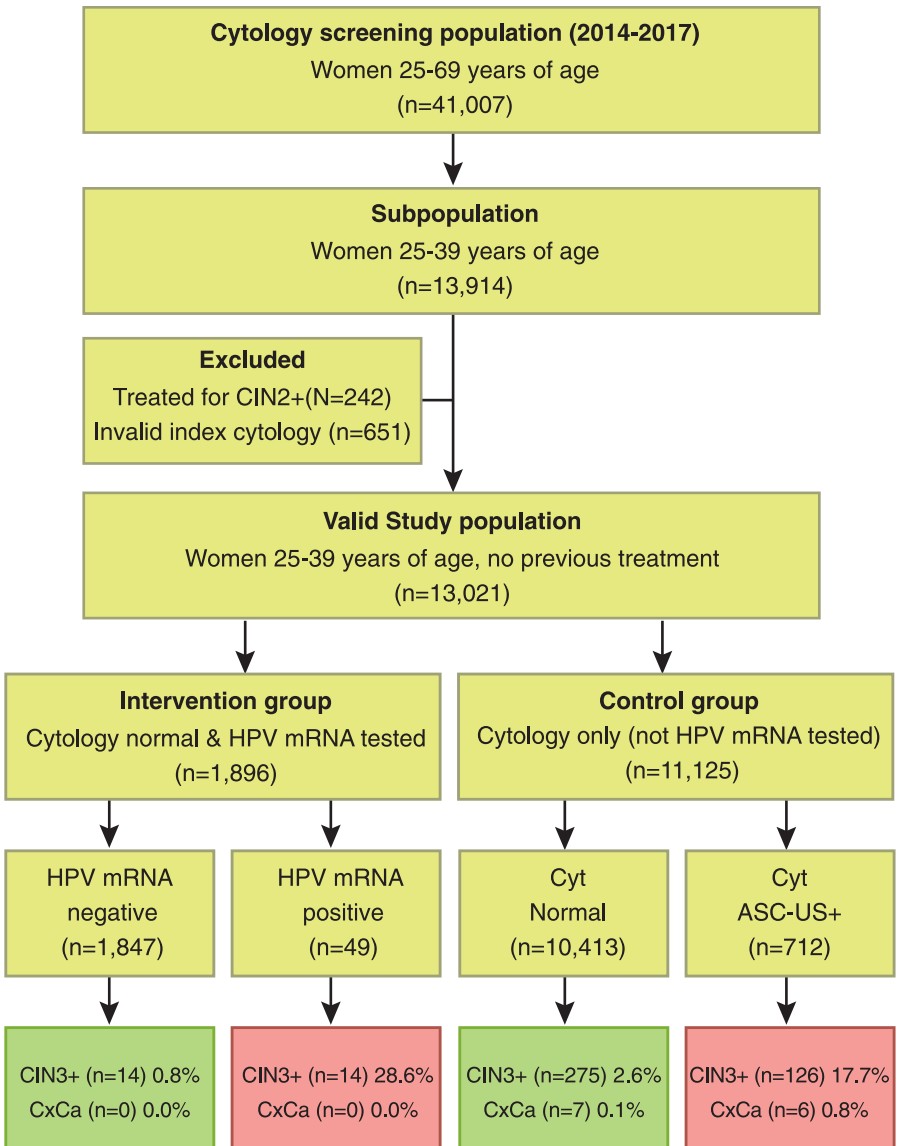

**Fig 1. Study population by cytology (ASC-US+), HPV mRNA and histology (CIN3+) for 5–7 years follow-up.**
CIN2+ = cervical intraepithelial neoplasia grade 2 (CIN2), CIN3, ACIS and cervical cancer. CIN3+ = CIN3, ACIS and cervical cancer. ASC-US+ = abnormal cytology (ASC-US, LSIL, ASC-H, HSIL, ACIS and cervical cancer). CxCa = Cervical cancer.

SPSS V.28 was used to conduct all statistical analyses, which entailed χ2 tests, Mann-Whitney tests and survival analyses (Kaplan-Meier estimator). A p value <0.05 was considered statistically significant.

## Ethical approval

The Regional Committee for Medical and Health Research Ethics (REC Nord) has approved the protocol as a quality assurance study in laboratory work (2013/497/REK Nord). Such studies are exempt from having a written informed consent from the patients.

## Results

Of the 13,021 women 25–39 years of age with index cytology in 2014–2017, 429 women (3.3%) had CIN3+ confirmed by biopsy in the follow-up through 2021, including 13 cases of invasive cervical cancer (14.3 per 100,000 women per year).

Of the 1,896 women with normal cytology and HPV mRNA test (intervention group), 49 women (2.6%) had a positive test. The cytology diagnoses were revised from normal to abnormal in 53.1% (26/49) of the HPV mRNA positive cases, while 23 cases remained normal after re-examination (true cytology negative, data not shown). The risks of CIN3+ among women with either positive or negative HPV mRNA test were 28.6% (14/49) versus 0.8% (14/1847), p<0.001 (Figs 1 and 2, S1 Table). The detection rates of HPV mRNA 16, 18 and 45 were 1.4% (26/1896), 0.5% (10/1896), 0.7% (13/1896), data not shown. The risks of CIN3+ in the follow-up for women having overexpression of HPV mRNA E6/E7 from types 16, 18 and 45 were 30.8% (8/26), 40.0% (4/10) and 15.4% (2/13), data not shown. None of the women in the intervention group developed cervical cancer during follow-up.

Of the 11,125 women with cytology only (control group), 712 women (6.4%) had abnormal cytology (ASC-US+). The risk of CIN3+ in women with abnormal and normal cytology were 17.7% (126/712) and 2.6% (275/10,413), p<0.001 (Figs 1 and, 2, S1 Table).

ASC-US+ = abnormal cytology (ASC-US, LSIL, ASC-H, HSIL, ACIS and cervical cancer) Using the Kaplan-Meier estimator, the cumulative incidence ratios (CIR) during seven years of follow-up for the four categories were: "Normal cytology / HPV mRNA negative" 0.8% (95% CI 0.4–1.2), "Normal cytology / Not HPV-tested" 2.7% (95% CI 2.4–3.0), "ASC-US+ / Not HPV-tested" 18.7% (95% CI 15.7–21.6) and "Normal cytology / HPV mRNA positive" 34.8% (95% CI 17.3–48.6) (Fig 3 and S1 Fig).

ASC-US+ = abnormal cytology (ASC-US, LSIL, ASC-H, HSIL, ACIS and cervical cancer) Of the fourteen CIN3+ with a negative 3-type HPV mRNA test at baseline, HPV testing (Cobas 4800) during follow-up identified two HPV 16, two HPV 18, five HPV "others" (not HPV 16/18), two HPV positive (not specified) and three women were not HPV-tested before the CIN3+ diagnosis. For seven of the fourteen women, the CIN3+ was detected 4–6 years after index cytology and might be a result of a new HPV-infection after baseline.

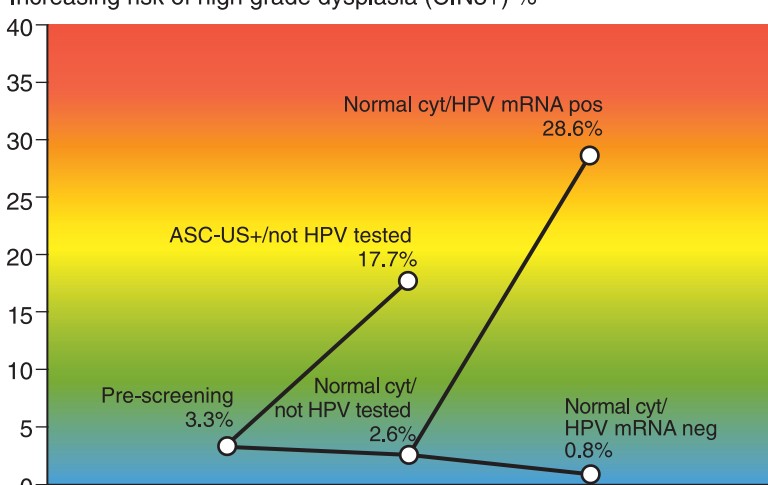

**Fig 2. The difference in risk of CIN3+ at specific branching points of the study.** CIN3+ = CIN3, ACIS and cervical cancer.

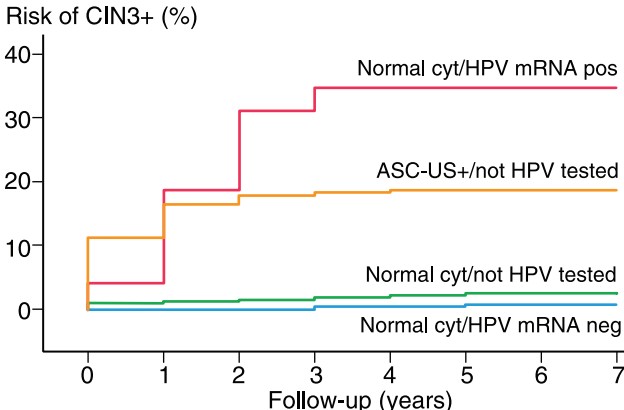

**Fig 3. Cumulative incidence ratio of CIN3+ for 7 years follow-up.** P < 0.001. CIN3+ = cervical intraepithelial neoplasia grade (CIN3), ACIS and cervical cancer. Normal cyt = Normal cytology = normal findings in screening of liquid-based cytology (LBC).

Of the 13 women (without an HPV mRNA test at baseline) who developed cervical cancer during the follow-up, 7 had normal and 6 had abnormal cytology at baseline. The risk of cervical cancer in women with normal cytology at baseline was 9.6 per 100,000 women per year.

All the six women with cervical cancer after an abnormal cytology (ASC-US+) at baseline had squamous cell carcinoma (SCC). Most of them got their cancer diagnosis a short time after baseline. Of the seven women with normal cytology at baseline there were five women with adenocarcinoma (ADC) and two women with SCC. Most of them were detected in the next screening round after three years (36 months), S2 Table. HPV testing (Cobas 4800) during follow-up identified five cases of HPV type 16 and one case of HPV type 18 as the causative agent among the cervical cancer incidents in the cytology control group. Seven women were not HPV DNA tested during follow-up. Two of the women with cervical cancer died in the study period, data not shown.

## Discussion

In this study, we observed that by testing women 25–39 years of age who have a normal cervix cytology result with a specific 3-type HPV mRNA test, an increase in screening programme sensitivity can be achieved without a substantial additional workload being imposed.

Cervical cancer is a preventable disease. The main cause of invasive cervical cancer is the deregulated and persistent production of HPV E6 and E7 oncoproteins [1]. Based on this fact, HPV E6/E7 mRNA could be a rational target for detecting the HPV infections that might lead to cellular transformation, offering advantages in preventing the harms of over screening and overtreatment that currently is challenging the management of patients, especially in younger women. This study found that only 2.6% (49/1,896) of the women aged 25–39 years showed an overexpression of HPV mRNA E6/E7, having negative cytology readings initially.

It is well known that cervical cytology is inherently limited in sensitivity and reproducibility due to being a subjective analysis method as opposed to molecular diagnostics which are objective [6, 10–13, 25, 26]. In the ATHENA study, the sensitivity of cytology varied from 42.0% to 73.0% [13]. In our study, out of 429 women with CIN3+ in the follow-up, only 29.4% (126/429) had abnormal cytology at baseline.

In Norway, the age standardised risk of cervical cancer was 12.3 per 100,000 women per year in 2021 (https://www.kreftregisteret.no/). In our study, the 7-year cumulative incidence

of cervical cancer was 14.3 per 100,000 women per year before screening and 9.6 per 100,000 women per year in women with normal cytology at baseline. These findings are in line with the reported incidence among normal cytology screening samples in the Netherlands, as described in the observational population based study by Rozemeijer et al. [27], In present material, there were no cases of cervical cancer in double negative women (Cyt-/HPV mRNA-), but by comparing the risk of CIN3+ in women with normal cytology (2.6%) versus the risk of CIN3+ in double negative women (0.8%), double test negative women are estimated to have a risk of cervical cancer of 3.0 per 100,000 women per year (9.6 per 100,000 x 0.8% / 2.6% = 3.0 per 100,000), using a 3-type HPV mRNA test. In a study from Kaiser Permanente Northern California (Berkeley, CA, USA), in 315,061 women, the 5-year cumulative incidence of cancer was 7.5 per 100,000 women per year for women with normal cytology, and 3.2 per 100,000 in double negative women, using the 13-type HPV DNA test Hybrid Capture II [26].

Reviewing the distribution of HPV mRNA genotypes towards histology diagnosis, we observed that HPV 16 and 18 were the most prevalent among CIN3+ cases, counting 30.8% (8/26) and 40.0% (4/10) followed by HPV 45 with 15.4% (2/13). This supports the findings by Froberg et al., showing that the presence of HPV 16 and 18 represent a significant risk of future cervical abnormalities needing to be treated, even if the cytology is negative at the time of infection. In the 9-year Swedish nested case-control follow-up study, Froberg et al. found that young women with normal cytology and a positive HPV-test for HPV16/18 need close follow-up. For women younger than 30 years of age, HPV type16/18 was significantly associated with the future risk of CIN2+, even though cytology was normal [28].

Norway has started, aiming to complete by the end of 2022, implementation of HPV DNA-based primary screening for cervical cancer for women 34–69 years old, with a more intense follow-up of the HPV 16/18 positives over the other 12 genotypes reported [29]. This will evidently improve screening sensitivity, while the risk of developing cervical cancer despite normal cytology remains unchanged for women 25–33 years old.

In the Nordic countries this age group (25–33 years old) carries the highest risk for CIN3+, where cervical cancer peaks at the age of 35 [30], highlighting the importance to focus on improved prevention for this specific age group. To prevent more cases of cervical cancer, effective quality assurance measures of cytology for the female subpopulation aged 25–33 years are investigated [14]. To tailor the most optimal algorithm for prevention, a trade-off between benefits (detected CIN3+), and harms (unnecessary colposcopies /biopsies) must be considered. Even though the benefits of a sensitive 14-type HPV DNA test are more numerous, the 3-type HPV mRNA test detecting oncogene activity from the most prevalent genotypes associated with cervical cancer makes a targeted quality control measure possible.

Several publications have compared the performance of different HPV tests in direct referral to colposcopy after an abnormal cytology diagnosis, ranging from mild dyskaryosis or worse [31, 32]. Sørbye et al. showed that the HPV mRNA test was more specific than DNA in triage of women with minor cervical lesions [33] and the work presented by Reinholdt et al. confirms a higher specificity of a 5-type HPV mRNA test especially among the younger women compared to a 14-type HPV mRNA test. "..the estimated specificity of the 14-type mRNA was low in women aged <40 years (10%-19%), indicating a potential high risk of unnecessary referrals in this age group [. . .]. The estimated specificity of the 5-type mRNA test was (64%-71%) respectively" [34].

Also St. Martin et al. investigated the outcome of different management strategies of low-grade cytology in young women below 30 years of age, comparing 5- and 14-type mRNA, 14-type DNA test, direct referral and repeat cytology. Their main findings were that HPV testing resulted in higher numbers of biopsies taken and that more women had insignificant findings which does not indicate treatment. "Except for the 5-type mRNA test, the relative risks for

biopsies with <CIN2 were higher than the relative risks for high grade lesions, indicating over management" [35].

In this study population the associated risk of CIN3+ among women positive for the 3-type HPV mRNA test was high (28.6%), even in this considered low-risk population having "normal" cytology readings prior to the re-examination of index cytology. Presented results are in line with the findings of Westre et al., where 32.2% (9/28) of the HPV mRNA positive / presumably cytology negative women were confirmed CIN3+ during follow-up [36].

Numerous studies have shown the long-term protection of HPV testing in women at risk of cervical cancer by comparing cumulative incidence ratio values between test combinations of HPV/Cytology.

Katki et al. [37] found that after an HPV positive/cytology negative test result, women had a 5-year CIN3+ risk at 4.5% (95% CI 4.2–4.8) [36], while Castel et al. reported CIR 2.8% (95% CI 2.1–3.7) respectively [38]. In another study, evaluating co-testing in a Spanish population of under screened women, the 5-year risk of CIN3+ among HPV positive/cytology negative women was reported to be 5.8% [39].

Our presented data (S1 Fig) show a 5-year CIN3+ risk at 34.7% in HPV mRNA 16, 18, 45 positive/cytology negative young women versus 0.8% in women being 3-type HPV mRNA negative/normal cytology. This supports the 3-type mRNA test in being a highly relevant biomarker for the identification of young women at higher risk and stresses the fact that the number of genotypes included in an HPV test carefully need to be balanced towards the corresponding low risk of developing CIN3+ observed for some of the genotypes such as type 39, 51, 56, 59 and 68. Commonly, young women below 40 years of age have higher number of CIN2+ cases, while the reported 5-year risk of CIN3+ is lower, reflecting the relative high number of CIN2 cases that regress if left untreated [40].

Finally, the additional workload of re-screening normal cytology cases which test positive with the 3-type HPV mRNA test is low, as only 2.6% cases in our study required rescreening. This volume is considered very low compared with other quality control measures currently used in cervical cytology, for example the use of different double screening methods like rapid pre-screening or rapid re-screening trying to reduce the risk of abnormal cells being misinterpreted or overlooked.

Inevitably, the suggested practice using mRNA HPV testing as a co-test to cytology negative samples in young women, to some extent increases the workload in the labs presumably as well as the cost of screening. However, molecular HPV-testing is considered less time-consuming compared to the manual evaluation of cytology slides in microscope. Research on how to further improve cytology readings seem quite limited, justified by the worldwide transition to a more sensitive HPV test in primary screening. Still, recent work making use of AI (Artificial Intelligence) reports that this might improve cytology readings [41–43], thus computer-aided screening might be an alternative to the 3-type HPV mRNA approach.

This study has not evaluated the cost-effectiveness of making use of a 3-type HPV mRNA test as a quality control measure of normal cytology, and detailed analysis must be done to verify suitability.

## Strengths and limitations

A major strength of this study is that it is population based, including all women participating in the Norwegian Cervical Cancer Screening Programme in Nordland County. The Department of Pathology at Nordlandssykehuset-Bodø receives all screening samples from this county. Overall, the coverage of the national screening program is 70% after three years and 80% after five years. As a result of the written personal invites to all women eligible for

screening, and the official reminders sent by the programme, close to 100% of the women screened during the study period 2014–2017 had a subsequent screen of follow-up within the end of 2021. Furthermore, all HPV mRNA test positive women at baseline have been followed-up with cytology and HPV-test after 12 months, even if the re-examination of index cytology resulted in a true negative result.

This study has some limitations, firstly the fact that only a small part of the study population (14.6%) has been HPV mRNA tested. Secondly, HPV mRNA testing was only performed at baseline, not in subsequent follow up nor testing of biopsies. Thirdly, due to compliance with national guidelines, cytology normal, HPV mRNA positive women were not referred to colposcopy and biopsy. This bias favouring cytology positive women to have a more thorough follow-up/increased surveillance might underestimate the performance of the mRNA test in our analysis. The lack of cost effectiveness studies to weigh the benefits of detecting more CIN3+ cases versus the added costs of molecular testing of normal cytology in young women is a drawback that needs to be pursued prior to implementation of such practice.

## Conclusions

By testing women 25–39 years of age with normal cytology with a specific 3-type HPV mRNA test, an increase in screening programme sensitivity can be achieved without an excessive additional workload. The volume of re-screened cytology samples is low (2.6%). The risk of CIN3 + among cytology normal, HPV mRNA E6/E7 types 16, 18, 45 positive women during follow-up is high (28.6%). The routine use of mRNA HPV test as an adjunct to cytology will identify women at an elevated risk, enabling a targeted quality control of cytology readings, thus improving programme sensitivity by early detection of cell abnormalities. When more women with CIN3+ are correctly identified and treated, less women will develop cervical cancer.

## Supporting information

**S1 Fig. Cumulative incidence ratio (CIR) of CIN3+ in the four groups for 5–7 years follow-up with confidence intervals (95% CI).**
(PDF)

**S1 Table. Category and risk of CIN3+ during follow-up (2014–2021).**
(PDF)

**S2 Table. Characteristics of cancer cases.**
(PDF)

**S1 File. Minimal dataset.**
(XLSX)

## Acknowledgments

We would like to extend our gratitude to the staff at the department of pathology at Nordlandssykehuset-Bodø for their kind co-operation, and to all the laboratory staff performing HPV-testing for their great work and collaboration during this study.

## Author Contributions

**Conceptualization:** Khalid Al-Shibli, Sveinung Wergeland Sørbye.

**Data curation:** Hiba Abdul Latif Mohammed, Ramona Maurseth, Mikkel Fostervold, Sebastian Werner.

**Formal analysis:** Khalid Al-Shibli, Mikkel Fostervold, Sveinung Wergeland Sørbye.

**Investigation:** Khalid Al-Shibli, Hiba Abdul Latif Mohammed, Ramona Maurseth, Sebastian Werner, Sveinung Wergeland Sørbye.

**Methodology:** Khalid Al-Shibli, Sebastian Werner, Sveinung Wergeland Sørbye.

**Project administration:** Khalid Al-Shibli, Mikkel Fostervold.

**Resources:** Khalid Al-Shibli.

**Validation:** Sveinung Wergeland Sørbye.

**Writing – original draft:** Khalid Al-Shibli, Sveinung Wergeland Sørbye.

**Writing – review & editing:** Khalid Al-Shibli, Hiba Abdul Latif Mohammed, Ramona Maurseth, Mikkel Fostervold, Sebastian Werner, Sveinung Wergeland Sørbye.

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
