## [Decision Letter · Decision Letter 0]

9 Aug 2022

PONE-D-22-122783-type HPV mRNA test in detection of CIN3+ in women 25-39 years with normal cervical cytologyPLOS ONE

Dear Dr. Sørbye,

Thank you for submitting your manuscript to PLOS ONE. After careful consideration, we feel that it has merit but does not fully meet PLOS ONE’s publication criteria as it currently stands. Therefore, we invite you to submit a revised version of the manuscript that addresses the points raised during the review process. The resubmitted manuscript appears more streamlined than before, however, readers unfamiliar with Norwegian screening program would struggle to understand some aspects of the work described. There are also some data discrepancies in the text that might be typos but could be confusing. Furthermore please aim to reduce the overlap of figures and tables shown. Essentially the same data is given 4 times plus the text. Personally table 1 seems most superfluous since Figure 1 contains info on the full population, figure 2 is quite illustrative (but needs a thorough legend) and figure 3 provides additional time data. Unless mRNA test can be done subsequently on the invasive cancer cases shown in Table 2 as suggested by Reviewer 2, this table could also be moved to the supplement material

We look forward to receiving your revised manuscript.

Kind regards,

Ivan Sabol

Academic Editor

PLOS ONE

Journal Requirements:

Additional Editor Comments:

The resubmitted manuscript appears to be much more streamlined version that now starts with a great numbed of cases and further elaborates a subset in more detail. This gives the readers a much broader view of the situation and presents a valuable dataset.

However, there are again some inconsistencies that are confusing for readers not familiar with the Norwegian screening program or where authors should more clearly present information.

For example P4L83 states there were 10724 women with cytology only however P6L130 says there were 11125. Since The table 1 is more in line with P6L130 apparently there is some error in the text at P4

P5 L106 only follow-up of women with positive mRNA test is described. It is uncertain how other women were followed up? When stating “Women with revised cytological diagnoses were followed-up in line with the Norwegian national guidelines until December 2021” introduces uncertainty since it is unknown how this revised cytological diagnosis was considered within this manuscript. The sentence should be revised to explain that all women were followed up and not only women with revised cytology.

Out of 49 cases with mRNA positive result, how many cases had revised diagnosis and was it a reexamination of the initial slide or collection of a new slide for mRNA positive women? How many cases were true cytology negative (remain negative) and how many were false negative (revised to abnormal) after revision? While actual grades (revision from normal to some stage of abnormal) might be irrelevant, some summary of how many times cytology performed sub optimally (if it is a reexamination of the existing slide) would be useful.

Further information on how follow up is actually done is needed. For example P6L138 implies the 3 years gap between testing, but figure 3 implies most CIN3+ events happened within 3 years even for cytology normal cases.

The text at P6 and Table 1 misuses the term PPV which often stands for positive predictive value; PPV = 100xTrue Positive/(True Positive +False Positive)). It is not a simple percentage

Table 1 and Figures 1, 2 and 3 essentially show the same data (0.8%, 2.6%, 17.7% and 28.6%). While it is clear that Figure 2 aims to visually show the difference in risk at specific branching points of the study, it could benefit from a more significant legend guiding the readers through the figure.

Reviewers' comments:

Reviewer's Responses to Questions

**Comments to the Author**

1. Is the manuscript technically sound, and do the data support the conclusions?

Reviewer #1: Yes

Reviewer #2: Yes

2. Has the statistical analysis been performed appropriately and rigorously? 

Reviewer #1: N/A

Reviewer #2: Yes

3. Have the authors made all data underlying the findings in their manuscript fully available?

Reviewer #1: Yes

Reviewer #2: Yes

4. Is the manuscript presented in an intelligible fashion and written in standard English?

Reviewer #1: Yes

Reviewer #2: Yes

5. Review Comments to the Author

Reviewer #1: This is an interesting paper from a study which examines whether using an mRNA HPV test following a cytology negative result in women <40 years screened as part of the Norwegian cervical cancer screening programme could increase screening sensitivity, identifying more women who are at risk of developing CIN3+.

However, not all Figures are needed or clearly presented and correctly labelled. The methods do not contain sufficient details about the screening programme and the follow-up of women in the study including the small group who receive a positive HPV result following normal cytology who have their sample re-examined. The strengths and limitations of the study are not discussed.

I hope that my queries and suggestions are helpful.

Title

I would suggest rewording the title. It doesn’t make sense grammatically or describe the study very clearly.

Introduction

Line 44-46. Consider changing to ‘HPV testing is performed for the early detection of cervical disease, both as a reflex test following cytology or as a primary screening method.’

Line 47. Specificity for what? i.e. for identifying women with high-grade lesions, CIN, cancer or something else?

Line 52. You say ‘by 2022’ but it is already mid-way through. Do you mean the end of 2022? Was it previously cytology based (I assume?)

Line 55. I think you mean that it’s not just that the screening programme hasn’t transitioned to primary HPV testing for this group but that cytology will remain the primary screening tool for this younger age group?

Line 58. It is normal practice to avoid starting a sentence with a number.

Line 59. I believe that terminology is moving away from using ‘smear’.

Line 62-64. Consider rephrasing this last sentence as it is crucial here that the reader understands your main point: i.e., ‘younger women are screened using cytology – given the high number of precancerous lesions in this group an mRNA reflex test following a negative cytology result could be used as a quality assurance measure to increase screening sensitivity’.

Line 66: there should be no apostrophe in HPVs.

Line 70: you use the word ‘proven’. Suggest rewording to ‘shown’ or ‘found’

Line 71: consider using a different word than ‘tool’

Line 72: ‘Elevated risk of future’. In line 62 you infer they are ‘high risk’ but here you state low-risk population.

Line 73: Consider rewording this for clarity. Also, avoid the use of ‘we’ but state as ‘the aim of this study was…’.

Methods:

I understand from the methods section that your baseline samples are taken as part of routine screening in 2014-2017. However, you do not state how long the interval is between routine screens for women of this age or what happens to this interval if they have a positive cytology result.

Line 108: Your main outcome measure is CIN3+. But I do not understand from the Methods how this outcome is measured, when and in whom? Everyone presumably? But as part of their subsequent screen – and if that’s the case, surely not everyone has a subsequent screen? Some must be lost to follow up? In the Results you mention ‘follow-up through 2021’ but you need to explain more in the Methods what the follow-up means.

Line 106: I think you are saying that in women with a positive mRNA HPV result their index cytology (originally deemed as ‘normal’) is re-examined by a technician. But you do not then present the numbers of how many are deemed as ‘normal’ when re-examined? What then happens to those with an ‘abnormal’ result on re-examination in terms of their screening internal? Colposcopy? What is their CIN3+ outcome? And what happens or would happen to those who are HPV+ve but cytology ‘normal’ on second reading, is the interval to their next screen shorter?

Results

Figure 1: You split by the group that have mRNA testing and the group that don’t. Have you considered splitting first by cytology normal and cytology abnormal, then splitting the cytology normal into those tested for HPV and those not – this makes a more intuitive tree – but ends up with the same four groups represented at the bottom? The diagram needs and title and footnotes to explain abbreviations.

Line 136: Correction ‘a short time’

Line 138: Correction ‘nest’ to ‘next’

Figure 2: I have no idea what this figure means. It doesn’t add anything to the results so consider removing.

Figure 3: Without knowing anything about the follow-up of these women, the screening cycles etc. this seems hard to interpret. The labeling seems insufficient, for example these four groups are actually 1) normal cytology + mRNA HPV positive 2) abnormal cytology (no HPV testing) 3) Normal cytology (no HPV testing) 4) normal cytology + mRNA HPV negative.

There is no info in the methods about how these data were obtained or what the graph represents so consider removing this or labeling better.

Table 1: See my comment about labeling of Figure 3. You do not mention this table in the Results. I do not think that the label PPV is correct. PPV in a clinical setting means what is the probability, given your positive result, that you really are positive. The cytology ‘abnormal’ result would be considered a ‘positive’ result as would the mRNA HPV positive group, but the other groups would be considered as a ‘negative’ result. I think what these values really are is the percentage who go on to develop CIN3+ within 7 years of the baseline result (i.e. 2014-2021).

Discussion

Line 173: why is this in quotations?

Line 177: consider using ‘intense’ rather than aggressive?

You do not discuss the strengths or limitations of the study. For example, the fact that there is no statistical analysis.

Using mRNA HPV testing as a reflex test is only one way to change the screening pathway for this age group, it might be appropriate to suggest alternatives (such as AI to improve cytology) or suggest future research e.g. a cost-effectiveness analysis.

I’m not clear on what you are suggesting happens to women who are found cytology normal mRNA HPV positive – colposcopy or a shorter screening interval?

Line 210: You say 2.6% of cases in our study needed to be rescreened – i.e. 2.6% of samples with normal cytology but positive mRNA HPV have their original sample re-evaluated by a technician. Using the word ‘rescreened’ suggests they provide a new sample, but I don’t think from the methods that this is the case.

Line 218: There is no excessive workload in terms of the number of samples that need to be re-examined but if HPV testing were used as a reflex test then every cytology normal sample would need a reflex HPV test and this would increase the workload in the labs presumably as well as the cost of screening.

Reviewer #2: General comments;

This study showed that three type HPV mRNA test increases the sensitivity in cervical cancer screening program without an excessive workload. This is interesting and important study to prevent cervical cancer incidence in young women. The authors also said that detection of three HPV types (HPV types 16, 18, and 45) in this assay might be best choice in the screening of women of 25-39 years of age, since some previous studies suggested that 90% of cervical cancer are positive with HPV16, 18 or 45 in this age group.

However, this reviewer does not convince with this hypothesis that this three-HPV mRNA test is the best choice for the above mentioned purpose, since this study showed that the sensitivity of CIN3+ in the intervention group was merely 50% (14/28), although the sensitivity was high enough as it was 97.4% (1847/1896). The authors are likely to say no cancer were found in these CIN3+ cases in the intervention group. However, to prove this assay is the best choice, the authors should show most of 13 cervical cancer cases in the control group in this study are positive by this assay. If possible, they also should examine what HPV types are positive in 14 CIN3+ cases showing negative with this HPV test in the intervention group. Such data may be important to support the findings.

6. PLOS authors have the option to publish the peer review history of their article (what does this mean?). If published, this will include your full peer review and any attached files.

Reviewer #1: No

Reviewer #2: **Yes: **Toshiyuki Sasagawa

---

## [Author Response · Author response to Decision Letter 0]

31 Aug 2022

Se cover letter and "Response to reviewers"

---

## [Decision Letter · Decision Letter 1]

26 Sep 2022

Impact of HPV mRNA types 16, 18, 45 detection on the risk of CIN3+ in young women with normal cervical cytology

PONE-D-22-12278R1

Dear Dr. Sørbye,

We’re pleased to inform you that your manuscript has been judged scientifically suitable for publication and will be formally accepted for publication once it meets all outstanding technical requirements.

Kind regards,

Ivan Sabol

Academic Editor

PLOS ONE

Additional Editor Comments (optional):

Reviewers' comments:

Reviewer's Responses to Questions

**Comments to the Author**

1. If the authors have adequately addressed your comments raised in a previous round of review and you feel that this manuscript is now acceptable for publication, you may indicate that here to bypass the “Comments to the Author” section, enter your conflict of interest statement in the “Confidential to Editor” section, and submit your "Accept" recommendation.

Reviewer #1: All comments have been addressed

2. Is the manuscript technically sound, and do the data support the conclusions?

Reviewer #1: Yes

3. Has the statistical analysis been performed appropriately and rigorously? 

Reviewer #1: Yes

4. Have the authors made all data underlying the findings in their manuscript fully available?

Reviewer #1: Yes

5. Is the manuscript presented in an intelligible fashion and written in standard English?

Reviewer #1: Yes

6. Review Comments to the Author

Reviewer #1: The authors have addressed each of the many points raised in the first review including removing unnecessary figures, including a strengths and limitations section and providing more details about the national screening programme.

7. PLOS authors have the option to publish the peer review history of their article (what does this mean?). If published, this will include your full peer review and any attached files.

Reviewer #1: **Yes: **Dr Susie Huntington

---

## [Editor Report · Acceptance letter]

9 Nov 2022

PONE-D-22-12278R1 

Impact of HPV mRNA types 16, 18, 45 detection on the risk of CIN3+ in young women with normal cervical cytology 

Dear Dr. Sørbye:

I'm pleased to inform you that your manuscript has been deemed suitable for publication in PLOS ONE. Congratulations! Your manuscript is now with our production department. 

Kind regards, 

on behalf of

Dr. Ivan Sabol 

Academic Editor

PLOS ONE